# Optimizing Knowledge Distillation in Transformers: Enabling Multi-Head Attention without Alignment Barriers

## Abstract

Knowledge distillation has been proven effective for compressing transformer architectures by transferring knowledge from teacher to student models. Logits-based methods of knowledge distillation cannot fully capture the intermediate representations and features within the teacher model, which may result in the student model not fully learning all the knowledge from the teacher model. Thus, previous work focuses on transferring knowledge through intermediate features or attention maps. However, leveraging multi-head attention maps in transformers for knowledge distillation presents challenges due to head misalignment and sub-optimal feature alignment, often requiring projectors to align features or special modifications to the model architecture. To address above limitations, we propose the Squeezing-Heads Distillation (SHD) method. This method reduces the number of attention maps to any desired number through linear approximation, without requiring additional projectors or parameters. This facilitates better alignment and knowledge transfer between models with different numbers of heads, enhancing both flexibility and efficiency. Experimental results demonstrate significant improvements in both language and vision generative models, validating the effectiveness of our method.

## 1 Introduction

In recent years, generative large models have experienced rapid growth, significantly impacting both NLP (e.g., GPT series[Brown (2020), Achiam et al. (2023)], LLaMA series[Touvron et al. (2023), Dubey et al. (2024)]) and computer vision domains (e.g., text-to-image generation[Esser et al. (2024)], text-to-video generation[Blattmann et al. (2023)]). Despite their impressive capabilities, these models typically involve a massive number of parameters, posing substantial challenges for practical online applications.

As the core of transformer models, the multi-head attention mechanism allows each head to attend to different parts of the input sequence, enabling the model to capture diverse and complex relationships between tokens. However, research such as Voita et al. (2019) has shown that only a small subset of attention heads significantly contributes to performance, suggesting redundancy among heads. By pruning the redundant heads, models can maintain their performance while reducing complexity. Similarly, Michel et al. (2019) demonstrates that most attention heads can be removed during testing without substantial performance degradation. Both studies and our observation3.1 suggest redundancy among multiple heads.

While pruning is a powerful compression technique, knowledge distillation offers another approach to model compression and performance improvement. Traditional knowledge distillation methods, particularly those developed for CNNs before the transformer era, focus on transferring knowledge through logits and intermediate feature maps. Techniques like DeiT[Touvron et al. (2021)] use a distillation token, and Patient-KD[Sun et al. (2019)] transfers intermediate features. However, transformers emphasize attention mechanisms, prompting research into distilling attention maps directly. TinyBERT[Jiao et al. (2019b)] and MobileBERT[Sun et al. (2020)] both explore the distillation of attention maps but require special model designs of matching head numbers between teacher and

student models, which is alignment barriers in knowledge distillation for transformers. See a more detailed description in our observation3.2.

We aim to design a practical method for generating multi-head attention map supervision from the teacher model during training, matching the student's head number to facilitate fine-grained knowledge transfer. This approach addresses head number mismatch, coarse-grained attention transfer, and the need for additional projectors. By introducing a non-standard attention matrix, we can achieve lossless feature representation with fewer heads. However, as stated in observation3.3, the unconstrained matrix lacks necessary attention knowledge for distillation.

In response to these challenges, we propose the **Squeezing Multi-Heads Distillation** method. This approach compresses multiple attention maps into a single attention map through efficient linear approximation, achieving fine-grained knowledge transfer between teacher and student models with different numbers of heads. The Squeezing Multi-Heads Distillation method offers several advantages. **Flexibility**: Unlike existing approaches that require matching attention head counts, our method allows for models with varying numbers of heads, broadening its applicability across different architectures. **Fine-grained Attention Knowledge**: it goes beyond simply transferring attention features $F \in \mathbb{R}^{N \times D}$ from the multi-head attention output, such as the Gram matrix $F^T F$, which provides a coarse approximation akin to single-head attention distillation. **Efficiency**: By compressing multiple attention maps into a single map using linear approximation, our method reduces computational overhead during distillation, improving overall efficiency compared with using traditional projectors with extra parameters.

The main contributions of this paper are as follows:

- We analyzed the behavior of multi-head compression using both unconstrained and constrained attention matrix approximations during training, and proposed an efficient Squeezing Multi-Heads Distillation(SHD) method based on linear approximation to address redundancy and alignment challenges in multi-head attention distillation.

- Our method provides a flexible and efficient solution for fine-grained knowledge transfer in knowledge distillation, which can be seamlessly integrated into existing distillation frameworks.

- We demonstrated the effectiveness of our method through comprehensive experiments on both language and vision generative tasks across diverse settings.

## 2 RELATED WORK

### 2.1 EFFICIENT MULTI-HEAD ATTENTION

The transformer architecture has revolutionized the field of natural language processing and computer vision, enabling the development of powerful models. One of the key components of the transformer is the attention mechanism, which allows the model to focus on relevant parts of the input sequence when making predictions. While inferencing these layers is often slow, due to the memory-bandwidth cost of repeatedly loading the large "keys" and "values" tensors. MQA[Shazeer (2019)] uses a single key-value head, drastically speeds up decoder inference, while GQA[Ainslie et al. (2023)] introduce grouped-query attention to avoid quality degradation of MQA. Research such as [Voita et al. (2019)] and [Michel et al. (2019)] conduct a detailed analysis of the functional roles of individual heads within the transformer's multi-head attention mechanism. Specifically, they assess whether certain heads are underperforming or redundant, and explore the feasibility of directly discarding these less contributive heads.

### 2.2 KNOWLEDGE DISTILLATION OF TRANSFORMER

Knowledge distillationHinton (2015) aims to train student networks by compressing or transferring knowledge from teacher model to student model. There are two common methods in this field, logits-based methods [Cho & Hariharan (2019), Furlanello et al. (2018), Mirzadeh et al. (2020), Zhang et al. (2018), Zhao et al. (2022)] which convey knowledge on the logits level and hint-based methods [Heo et al. (2019), Huang & Wang (2017), Kim et al. (2018), Park et al. (2019), Peng et al. (2019)] which convey knowledge through intermediate features. As an example of using both

above methods in knowledge distillation of transformer, DistillBERT(Sanh et al. (2019)) initializes the student with teacher's partial parameters, and minimized the soft target probabilities and cosine similarity of hidden states between the teacher and the student. Through Alishahi et al. (2019) find that the attention weights learned by BERT can capture substantial linguistic knowledge, Tiny-BERT(Jiao et al. (2019a)) propose the attention based distillation to encourage that the linguistic knowledge can be transferred from teacher to student. MobileBERT(Sun et al. (2020)) train a specially designed inverted-bottleneck and bottleneck structures to keep their layer number and hidden size the same for the teacher and the student, transferring knowledge through feature maps and self-attention maps. MINILM(Wang et al. (2020)) introduce the scaled dot-product between values in the self-attention module as the new deep self-attention knowledge, in addition to the attention distributions. However, the above work require that the number of attention heads must be same for the teacher and the student, which is not in line with reality.

## 3 OBSERVATIONS

### 3.1 OBSERVATION 1: HEAD REDUNDANCY IN TRANSFORMERS

Multi-head attention is a very common technique that improves the performance of attention mechanisms in transformer models, leading to significant improvements. However, we observed that different heads often capture similar or redundant attention pattern, as illustrated in Fig.1.

The most common patterns of attention maps are diagonal and vertical lines, indicating the importance of adjacent tokens or key elements. This phenomenon is prevalent in well-trained generative models, especially the large ones with many heads. This suggests that the number of heads is redundant to some degree.

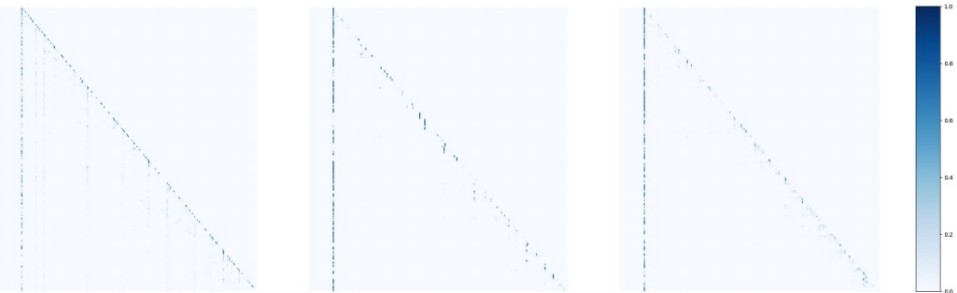

Figure 1: Attention maps during inference on a random sample in Dolly Datatset by GPT2-XL from different heads of one layer. Each column is from the same head. We random selected three heads to visualize it. The attention patterns are very similar and contain much redundancy within one layer. Only Response attentions are kept and Instruction attentions are masked.

### 3.2 OBSERVATION 2: ALIGNMENT BARRIERS IN KNOWLEDGE DISTILLATION FOR TRANSFORMERS

In recent literature, we have identified two major alignment barriers in knowledge distillation for transformers.

First, the significant dimensional gap between large and small transformer models makes feature alignment less effective. Knowledge distillation in generative models faces challenges due to differences in hidden dimensions; large models typically have higher-dimensional hidden layers, making feature alignment with smaller models difficult. For example, GPT-3[Brown (2020)]'s model dimension is 12,288, which is almost 20 times larger than that of GPT-3[Brown (2020)] Small. Some methods introduce projection layers to align features[ Sun et al. (2020)], but these methods add extra parameters and do not focus on the attention module. Consequently, most feature alignment knowledge distillation methods appear ineffective for generative models.

Second, transformer models often differ in the number of attention heads, leading to head alignment problems. Previous methods like MobileBERT[ Sun et al. (2020)] employed specialized model designs to circumvent the head alignment issue, but this required both teacher and student models to be specifically engineered, potentially reducing performance. To our knowledge, we are the first to directly address the head alignment problem of attention maps in knowledge distillation.

### 3.3 Observation 3: Rank Limitations of Attention Maps and Their Implications

Consider a self-attention module without the causal mask, as used in many diffusion models. The hidden dimension per head ($d$) is often much smaller than the number of tokens ($N$). For example, DiT models have $d = 64$ per head but are trained on images with $64 \times 64 = 4096$ tokens. This also occurs in most generative models.

Reviewing how the attention map of a single head is computed:

$$A_i = \mathrm{softmax}\left(\frac{Q^i (K^i)^\top}{\sqrt{d}}\right), \tag{1}$$

where $A_i \in \mathbb{R}^{N \times N}$ is the attention map of the $i$-th head, $Q^i, K^i \in \mathbb{R}^{N \times d}$ are the queries and keys, $N$ is the number of tokens, and $d$ is the hidden dimension per head.

Since $N > d$ in most cases, the rank of $A_i$ is limited to at most $d$, making $A_i$ a rank-deficient matrix. As a result, the attention map of a single head cannot fully utilize its theoretical capacity, leading to redundancy from a linear algebra perspective.

To explore this further, consider introducing a non-standard attention matrix $\tilde{A} \in \mathbb{R}^{N \times N}$, which is not constrained by non-negativity or the requirement that each row sums to one. By allowing all heads to replace the standard attention with a shared $\tilde{A}$, we can achieve lossless feature representation with just a single head, as the matrix offers $N^2$ degrees of freedom but is constrained to only $N \times d$. This provides a mathematical basis for head compression.

However, the unconstrained nature of $\tilde{A}$ lacks the structured attention knowledge required for tasks like distillation. In the following sections, we will propose a efficient solution to incorporate more meaningful knowledge into this representation while maintaining a reasonable feature representation loss.

## 4 Squeezing Heads: A Better Supervision Beyond Just Logits

Traditional knowledge distillation algorithms primarily focus on feature distillation. However, the gap in hidden dimensions between teacher and student transformer models makes feature alignment challenging. Attention maps provide a better supervision signal since they are independent of hidden dimensions. Yet, teachers and students often have different numbers of heads due to computational constraints in model design. Our method addresses this issue by improving knowledge transfer. We propose a technique that compresses diverse multi-head attention into a single attention map while preserving key information, which can then be used as supervision for knowledge distillation. Initially, we introduce lossless unconstrained attention compression and optimal constrained attention compression based on teacher features, but these methods are computationally inefficient for every training iteration. We then present our "squeezing heads" method, which compresses attention maps by efficient linear approximation and achieves strong results in practice.

### 4.1 Attention compression by Exact Optimization

Let's examine multi-head attention more closely. Generally, attention consists of two parts: (1) Scaled Dot-Product Attention and (2) Multi-Head Attention. The scaled dot-product attention is defined as:

$$\mathrm{Attention}(Q, K, V) = \mathrm{softmax}\left(\frac{QK^\top}{\sqrt{d}}\right) V. \tag{2}$$

The multi-head attention mechanism can be described as:

$$\text{MultiHead}(Q, K, V) = \text{Concat}(\text{Head}_1, \ldots, \text{Head}_h)W^O,$$
$$\text{where } \text{Head}_i = \text{Attention}(QW_i^Q, KW_i^K, VW_i^V), \tag{3}$$

where $W_i^Q, W_i^K, W_i^V \in \mathbb{R}^{d_{\text{model}} \times d}$, and $W^O \in \mathbb{R}^{hd \times d_{\text{model}}}$ are projection matrices.

To better understand our method, we can expand $W^O$ as concatenated per-head output projections $W_i^O \in \mathbb{R}^{d \times d_{\text{model}}}$[Elhage et al. (2021)]. Then, the multi-head attention can be rewritten as:

$$\text{MultiHead}(Q, K, V) = \sum_{i=1}^h \text{Head}_i W_i^O = \sum_{i=1}^h A_i V W_i^V W_i^O,$$
$$\text{where } A_i = \text{softmax}\left(\frac{QW_i^Q(KW_i^K)^\top}{\sqrt{d}}\right). \tag{4}$$

From an information theory perspective, we try to compress multiple attention maps into a single attention map while preserving feature representation. We denote $VW_i^V W_i^O$ as $X_i$ for simplicity of symbols. Consider combining two attention heads $A_{2i-1}$ and $A_{2i}$ into a single attention map $\tilde{A}_i$ that satisfies:

$$\tilde{A}_i = \arg\min_{\tilde{A}_i} \left\| \tilde{A}_i \left( X_{2i-1} + X_{2i} \right) - \left( A_{2i-1} X_{2i-1} + A_{2i} X_{2i} \right) \right\|_F^2.$$

This unconstrained optimization problem seeks the $\tilde{A}_i$ that best aligns the combined transformed value vectors before and after compression, effectively "squeezing" the attention heads by finding the optimal aggregate attention map $\tilde{A}_i$.

We observe that the loss can actually be reduced to zero, providing a closed-form solution for $\tilde{A}_i$ regardless of how many heads are compressed, due to the additional degrees of freedom. However, directly computing $\tilde{A}_i$ using the pseudo-inverse is impractical during training because it has a computational complexity of $O(N^6)$ and this solution does not consider the constrain of the attention matrix and can not provide token relation knowledge.

If we constrain the optimization problem by restricting the values in $\tilde{A}_i$ to the range $[0, 1]$ and ensuring that each row sums to 1, the problem becomes convex with a global minimum solution. Nonetheless, the computational complexity remains $O(N^6)$, making it equally impractical to apply during training.

## 4.2 Linear Approximation for Aligning Attention Maps

Obtaining exact attention compression results is computationally expensive. To mitigate this, we propose a linear combination of attention maps to approximate the combined effect. For simplicity, we demonstrate this by squeezing two heads into one; however, our method can be easily extended to compress any number of heads into one. We reparameterize $\tilde{A}_i$ as a linear combination of known attention maps as follows:

$$\tilde{A}_i = \alpha_i A_{2i-1} + (1 - \alpha_i) A_{2i}, \tag{5}$$

where $\alpha_i \in [0, 1]$ is a scalar weight to be determined.

Our goal is to find $\alpha_i$ that minimizes the difference between the combined output using $\tilde{A}_i$ and the original outputs:

$$\alpha_i = \arg\min_{\alpha} \left\| \tilde{A}_i(X_{2i-1} + X_{2i}) - (A_{2i-1} X_{2i-1} + A_{2i} X_{2i}) \right\|_F^2, \tag{6}$$

where $X_{2i-1} = VW^V_{2i-1}W^O_{2i-1}$ and $X_{2i} = VW^V_{2i}W^O_{2i}$.

Expanding the expression, we have:

$$E(\alpha_i) = \|(\alpha_i A_{2i-1} + (1-\alpha_i)A_{2i})(X_{2i-1} + X_{2i}) - (A_{2i-1}X_{2i-1} + A_{2i}X_{2i})\|^2_F$$
$$= \|\alpha_i M + N\|^2_F, \tag{7}$$

where we denote $M = (A_{2i-1} - A_{2i})(X_{2i-1} + X_{2i})$, $N = A_{2i}X_{2i-1} - A_{2i-1}X_{2i}$ for simplicity.

To find the optimal $\alpha_i$, we set the derivative of $E(\alpha_i)$ with respect to $\alpha_i$ to zero:

$$\frac{dE(\alpha_i)}{d\alpha_i} = 2\alpha_i\|M\|^2_F + 2\langle M, N\rangle = 0, \tag{8}$$

where $\langle M, N\rangle$ denotes the Frobenius inner product.

Solving for $\alpha_i$, we get:

$$\alpha_i = -\frac{\langle M, N\rangle}{\|M\|^2_F}. \tag{9}$$

Since $A_{2i-1}$ and $A_{2i}$ are attention maps with non-negative elements and row sums equal to 1, and $\alpha_i$ is computed to minimize the reconstruction error, in practice $\alpha_i$ often falls within the interval $[0, 1]$.

By using this linear combination, we can effectively squeeze the attention maps from two heads into one, aligning the teacher's attention maps with those of the student model that has fewer heads, thus facilitating better knowledge transfer during distillation. Progressively we can merge heads into arbitrary numbers.

### 4.3 TRAINING OBJECTIVE OF SQUEEZING HEADS DISTILLATION

Attention maps represent discrete categorical distributions. Inspired by logit distillation, we introduce attention temperature to enhance low-probability regions. Consequently, we modify Eq.10 to incorporate attention temperature:

$$A_i = softmax(\frac{Q^i(K^i)^\top}{\sqrt{d}T_a}) \tag{10}$$

where $T_a$ is the attention temperature which is manually set and normally larger than 1.0. The function of attention temperature is the same as temperature in logit distillation, which is to soften the output probabilities and make it more uniform, highlighting the relative differences between tokens more clearly.

Since our method only focuses on the supervised attention maps of teachers, it can be plugged into all logit-based distillation. The Squeezing Heads Distillation Loss is expressed as:

$$L_{SHD} = \beta \sum_{i=1}^{L} \sum_{j=1}^{H_s} L_{KL}(\tilde{A}^t_i, A^s_i) \tag{11}$$

where $L_{KL}$ is the Kullback-Leibler divergence loss. We add it to the original training loss in our experiments with $beta$ to control the intensity.

## 5 EXPERIMENTS

To validate the improvement of our method, we performed several experiments on different generative transformer models. We trained diffusion transformer models for image generation tasks and trained various LLMs for both LLM pretraining tasks and supervised fine-tuning tasks. All the LLM and diffusion generative models selected are transformer models. We apply our method to

self-attention blocks of all student model layers. We select image generation tasks and LLM tasks because those two represent generative tasks nowadays since diffusion transformer models use full attention on the whole image or latent space and LLMs use causal attention to performance in an auto-regressive manner. Those two represent the most generative methods in two major AIGC fields.

Suppose teacher and student models have different layers of transformer block. In that case, we select the corresponding layers of the teacher model for the student, e.g. if the teacher has 48 layers and the student has 24 layers, the supervision of student's $4^{th}$ layer will be the teacher's $8^{th}$ layer.

## 5.1 MAJOR RESULTS

| Method | Params | Model | Image Res | Steps | FID-50K↓ | IS↑ | Prec↑ | Rec↑ |
|---|---|---|---|---|---|---|---|---|
| teacher | 130M | MDTv2-B/2 | 256x256 | 400k | 35.77 | 54.01 | 0.48 | 0.62 |
| w/o KD | 33M | MDTv2-S/2 | 256x256 | 400k | 44.87 | 37.29 | 0.47 | **0.49** |
| KD | 33M | MDTv2-S/2 | 256x256 | 400k | 38.73 | 43.43 | 0.50 | 0.48 |
| KD+SHD | 33M | MDTv2-S/2 | 256x256 | 400k | **36.95** | **46.27** | **0.52** | 0.48 |
| teacher | 675M | MDTv2-XL/2 | 256x256 | 3500k | 1.58 | 314.73 | 0.79 | 0.65 |
| w/o KD | 33M | MDTv2-S/2 | 256x256 | 500k | 42.33 | 40.38 | 0.48 | 0.49 |
| KD | 33M | MDTv2-S/2 | 256x256 | 500k | 33.08 | 49.23 | 0.52 | 0.57 |
| KD+SHD | 33M | MDTv2-S/2 | 256x256 | 500k | **32.27** | **50.54** | **0.53** | **0.57** |

Table 1: Performance comparison for image generation on ImageNet-1K. The detailed definitions of metrics Prec and Rec can be found in Kynkäänniemi et al. (2019).

**Image Generation.** We did several experiments on MDTv2 models under different model sizes, as shown in 1. We select MDTv2-S/2-MDTv2-XL/2, and MDTv2-S/2-MDTv2-B/2 as the student-teacher pairs for different model sizes. With simple KD, our methods can improve FID and IS scores by a huge margin on both settings, achieving a 36.95 FID score when training 400k steps for MDTv2-S/2 model. It shows that with different teacher model size and large capacity gaps, our method still benefits the student model's representation learning.

**Pretraining on LLM.** The results after fine-tuning are reported in Tab.2. We can observe that extremely small models like BabyLLaMA (58M) still can gain improvement from our method in this experiment. The performance of our method beats models of twice our size on several evaluation datasets like SST-2, MRPC, QQP and MNLI-mm, while performance on other datasets also improved compared to our baseline. We can see that our method to imitate attention maps in the pretraining shows great generalization capability on downstream tasks.

| Model | OPT(base) | T5(base) | BabyLLaMA | BabyLLaMA+SHD |
|---|---|---|---|---|
| Size | 125M | 222M | 58M | 58M |
| CoLA(MCC) | 15.2 | 11.3 | 15.6 | **17.5**(+1.9) |
| SST-2 | 81.9 | 78.1 | 85.8 | **88.4**(+2.6) |
| MRPC(F1) | 72.5 | 80.5 | 81.6 | **82.0**(+0.4) |
| QQP(F1) | 60.4 | 66.2 | 82.8 | **83.1**(+0.3) |
| MNLI | 57.6 | 48.0 | **72.9** | 72.8(-0.1) |
| MNLI-mm | 60.0 | 50.3 | 73.7 | **74.0**(+0.3) |
| RTE | 60.0 | 49.4 | 58.6 | **58.6**(+0.0) |
| BoolQ | 63.3 | 66.0 | 59.8 | **61.7**(+1.9) |
| MultiRC | 55.2 | 47.1 | 54.6 | **59.0**(+4.4) |
| WSC | 60.2 | 61.4 | 53.0 | **56.6**(+3.3) |

Table 2: Fine-tuning accuracy (if not specified), MCC score or F1 score evaluated by SuperGLUE on language pretraining task.

**Supervised Fine-tuning on LLM.** We evaluated our method in Tab.3. Our method beats our baseline MiniLLM[Gu et al. (2024)] by 0.8% on DollyEval. This is the major metric we look on since the models are trained on Dolly. Our model also gains huge improvement on other test sets like S-NI UnNI, achieving SoTA in the same model and training settings. The experiment on SelfInst also shows that our method can benefit the student's representations even when the student is better than the teacher's performance (14.3->15.2). The knowledge transferred by SHD is the ability to model long-range dependencies. The student can always benefit from a model with more parameters, even if it is not optimized to its full state (in our experiment, the teacher only trained with SFT). We also show our method over three different student model sizes. We have made almost all improvements except on S-NI with the 760M student model. The teacher is not well trained compared to the student in this case. On relatively modern and large models like LLaMA-13B and LLaMA-7B, our SHD still can boost the performance, gaining 1.1% improvement on UnNI.

| Method | Head | Params | DollyEval | SelfInst | VincunaEval | S-NI | UnNI |
|---|---|---|---|---|---|---|---|
| Teacher(GPT2-XL) | 25 | 1.5B | 27.6 | 14.3 | 16.3 | 27.6 | 31.8 |
| SFT w/o KD | 12 | | 23.3 | 10.0 | 14.7 | 18.5 | 18.5 |
| KD | 12 | 120M | 22.8 | 10.8 | 13.4 | 16.4 | 22.0 |
| MiniLLM | 12 | | 24.6 | 13.2 | 16.9 | **25.1** | 25.6 |
| MiniLLM+SHD | 12 | | **24.8** | **13.6** | **18.0** | **25.1** | **25.7** |
| SFT w/o KD | 16 | | 25.5 | 13.0 | 16.0 | 25.1 | 32.0 |
| KD | 16 | 340M | 25.0 | 12.0 | 15.4 | 23.7 | 31.0 |
| MiniLLM | 16 | | 25.4 | 14.6 | **17.7** | 27.4 | 31.3 |
| MiniLLM+SHD | 16 | | **26.2** | **15.2** | **17.7** | **28.1** | **32.2** |
| SFT w/o KD | 20 | | 25.4 | 12.4 | 16.1 | 21.5 | 27.1 |
| KD | 20 | 760M | 25.9 | 13.4 | 16.9 | 25.3 | 31.7 |
| MiniLLM | 20 | | 26.4 | 15.9 | 17.7 | **29.2** | 33.0 |
| MiniLLM+SHD | 20 | | **26.5** | **16.2** | **18.2** | 28.9 | **33.5** |
| Teacher(LLaMA-13B) | 40 | 13B | 29.7 | 23.4 | 19.4 | 35.8 | 38.5 |
| MiniLLM | 32 | 7B | 28.9 | 23.1 | 19.4 | 34.8 | 37.4 |
| MiniLLM+SHD | 32 | | **29.1** | **23.4** | **20.0** | **34.9** | **38.5** |

Table 3: Performance Comparison of distillation of LLM on various Test sets, supervised fine-tuning on Dolly.

**Image Classification on ImageNet-1k.** To show the effectiveness of our methods, we also did image classification to prove our method can be applied to discriminative tasks. We followed the original settings of ViTKD[Yang et al. (2022)] and NKD[Yang et al. (2023)], which focuses on knowledge distillation of ViT-ViT teacher-student training pairs. All experiments are conducted on ImageNet-1k in the Table 4. The teacher model is DeiT3-small and the student model is Deit-Tiny. The result of ViTKD+NKD+SHD also shows the compatibility of SHD with FD methods, improving the performance of ViTKD+NKD by 0.42% on a strong baseline.

| Method | Model | Head | Epochs | Top1 Acc |
|---|---|---|---|---|
| Teacher | DeiT3-small | 6 | 300 | 80.69 |
| Baseline (without KD) | DeiT-Tiny | 3 | 300 | 74.43 |
| Baseline+SHD | DeiT-Tiny | 3 | 300 | **75.38** |
| ViTKD+NKD | DeiT-Tiny | 3 | 300 | 77.79 |
| ViTKD+NKD+SHD | DeiT-Tiny | 3 | 300 | **78.21** |

Table 4: Performance Comparison for image classification on ImageNet-1K.

## 5.2 ABLATIONS AND ANALYSIS

**Comparison with other representation distillation methods.** Traditional Knowledge distillations always solve the dimension-alignment problem by using projectors or distilling the relations between features which is called Feature Distillation (FD). The projector aligns the student's feature to the teacher's dimension by a single linear projection or an MLP, causing more training parameters to train. The other previous works distill the relations of the features among tokens or spatial-wise pixels. The most common way is to calculate the self-correlations:

$$Cor = \frac{FF^T}{\|F\|\|F\|}, L_{cor} = 1 - Sim(Cor^t, Cor^s) \tag{12}$$

where $F$ is the intermediate feature and $Sim$ is any similarity score measured by some function. We did comparison experiments in Tab.5. We did hyperparameters searching like our method did and their best results are reported. The details of training speed, training parameters, and FLOPs are in the Appendix. We also compared one of the SoTA FD methods: VkD[Miles et al. (2024)]. We also did an ablation study in which we used similarities of attention maps between all heads and selected the pairs of heads of maximum similarity to merge heads on training set before training. We call it "head_matching" in the Table 5. Our method surpasses those two methods on all evaluation datasets except SelfInst results, which are the same, without extra training costs.

| Method | DollyEval | SelfInst | VincunaEval | S-NI | UnNI |
|---|---|---|---|---|---|
| MiniLLM | 25.4 | 14.6 | **17.7** | 27.4 | 31.3 |
| MiniLLM+FD+Projector | 25.8 | **15.2** | 17.6 | 27.3 | 31.4 |
| MiniLLM+FD+self_correlation | 25.9 | **15.2** | 15.8 | 26.8 | 31.7 |
| MiniLLM+VkD | 26.0 | 14.9 | **17.7** | 27.1 | 31.0 |
| MiniLLM+SHD | **26.2** | **15.2** | **17.7** | **28.1** | **32.2** |
| MiniLLM+SHD+head_matching | **26.3** | **15.3** | **18.2** | **28.1** | **32.3** |

Table 5: Performance Comparison with other representation distillation methods.

We show that forcing students to only imitate the relations or the features can cause bad harm to model performance as well, resulting in a degradation of VincunaEval and S-Ni datasets, while our method can benefit from attention relations. This is also proven mathematically by our method part. Our method has the lowest loss of teachers knowledge transfer in the measurement of output.

**Hyperparameters.** The results of different hyperparameters are listed in Tab.6 on image generation. Our method is not very sensitive to hyperparameters and does not require many maunal settings. Also, the attention temperature we propose can improve the results as well.

| temperature | beta | FID | IS | precision | Recall |
|---|---|---|---|---|---|
| 1 | 0.5 | 37.73 | 45.25 | 0.51 | 0.49 |
| 1 | 2.0 | 38.64 | 44.16 | 0.50 | 0.48 |
| 1 | 5.0 | 39.39 | 43.86 | 0.49 | 0.49 |
| 2 | 2.0 | 36.95 | 46.27 | 0.51 | 0.48 |

Table 6: Performance comparison for different hyper-parameters for image generation on ImageNet-1K for SHD.

**Is SHD better than hard selecting on heads?** Hard selecting is another way to squeeze heads from teacher to student. We did an experiment in which we randomly selected $H^s$ heads from teacher models before training and used the attention maps of those heads to supervise the student's training. The results are reported in Tab.7 on image generation. The performance dropped drastically with hard selecting several heads compared to KD baselines. This shows that the dropping heads of teachers can be very important for students. The capacity gap from teacher to student can not be ignored for hard selecting.

| | FID | IS | precision | Recall |
|---|---|---|---|---|
| w/o KD | 42.33 | 40.38 | 0.48 | **0.49** |
| KD | 38.73 | 43.43 | 0.50 | 0.48 |
| KD+hard select | 40.03 | 42.22 | 0.50 | 0.48 |
| KD+SHD | **36.95** | **46.27** | **0.52** | 0.48 |

Table 7: The performance comparison on ImageNet-1K for Image Generation of selecting different attention maps source.

This experiment also indicates the priority of our method over hard selection. If only part of the teacher knowledge has been transferred among heads, it can be harmful to the student. Our method of soft merging is a better option.

**Is SHD better than constant merging on heads?** SHD always calculates sample-wise $\alpha$ for different heads and layers. A naive thought is to merge the teacher's heads by simple constant $\alpha = 0.5$. We also did ablation studies for this experiment in Tab.8. It achieves comparable results with the original MiniLLM baseline, but also cannot improve the performance like SHD does. SHD is sample-wise and more fine-grained.

| Method | DollyEval | SelfInst | VincunaEval | S-NI | UnNI |
|---|---|---|---|---|---|
| MiniLLM | 25.4 | 14.6 | **17.7** | 27.4 | 31.3 |
| MiniLLM+constant merge | 25.5 | 15.0 | **17.7** | 27.1 | 31.5 |
| MiniLLM+SHD | **26.2** | **15.2** | **17.7** | **28.1** | **32.2** |

Table 8: Comparison of merging heads. "Constant merge" indicates a combination of the teacher's attention maps via different heads in a still constant manner. Our method is sample-wised.

**Does SHD works independently?** We did this ablation study on discriminative tasks in Table.4. We used SHD independently and the result still improved 0.95% without logits KD.

## 6 CONCLUSION

In this work, we introduce the Squeezing-Heads Distillation (SHD) method, a novel approach to knowledge distillation that addresses the challenges posed by head misalignment and redundancy in multi-head attention mechanisms of transformer models. Our method effectively compresses multiple attention maps into a single map through a linear search process, enabling better alignment and knowledge transfer between models with different numbers of heads. This approach not only enhances the flexibility and efficiency of the distillation process but also improves the performance of student models across various generative tasks.

We validate the effectiveness of SHD through comprehensive experiments on both image generation and language pretraining tasks. Our results demonstrate significant improvements in key metrics such as FID, IS, and accuracy, outperforming traditional knowledge distillation methods and other representation distillation techniques. The proposed method also proves to be robust across different model sizes and architectures, showing its general applicability.

Furthermore, our ablation studies confirm that SHD provides a more stable and efficient improvement compared to hard selecting or constant merging of heads. The introduction of attention temperature further enhances the distillation process by softening the output probabilities, leading to better performance in student models. Notably, SHD achieves these improvements without introducing additional training parameters or significantly impacting training speed.

In conclusion, the Squeezing-Heads Distillation method offers a practical and effective solution for optimizing knowledge distillation in transformers, enabling the deployment of smaller, more efficient models without compromising on performance. This work paves the way for further research into flexible and efficient distillation techniques that can adapt to the diverse architectures of modern large-scale models.

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

# A  APPENDIX

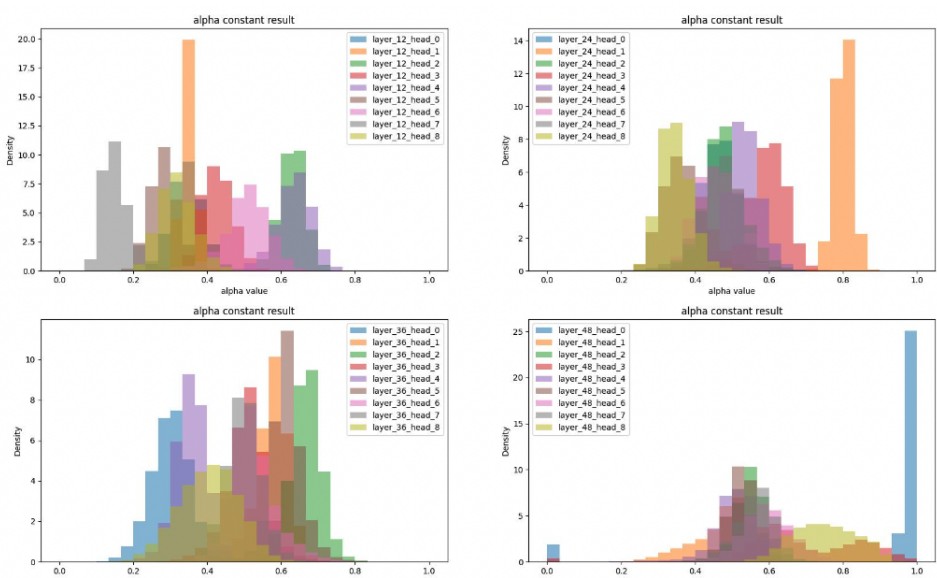

Figure 2: Distribution of $\alpha$ from different layers of GPT2-XL. Same color represents same head.

**Training details.** For image generation tasks, we used the families of MDTv2 architecture which is a variant of DiT models. We use it to prove that our method is compatible with different kinds of methods. We used the MDT-B/2 model as the teacher model and MDT-S/2 model as the student model, which are both trained on ImageNet-1k with a resolution of 256x256, using a 256 batch size and Adan optimizer. Other settings are also aligned with the original MDT and DiT. The $\beta$ we set on image generation tasks is 2.0.

For language pretraining tasks, we trained LLaMA models on the BabyLM dataset. We compare our method with BabyLLaMA. BabyLLaMA averaged two teacher model logits as an ensembled teacher. The teacher models used are GPT-2 and LLaMA. Experiments are conducted only on one GPU. The models are trained for 6 epochs with a batch size of 256 and for a learning rate of $2.5 \times 10^{-4}$. We take one teacher's attention map for squeezing head distillation. We followed all the settings of BabyLLaMA. The $\beta$ we set is 1.0. We retrained BabyLLaMA with BabyLM with the original settings and official code and reevaluated the metrics.

For supervised fine-tuning tasks, we follow the setting of MiniLLM. we randomly selected 12500 samples for training, 1000 samples for validation, and 500 samples for testing from databricks-dolly-15k dataset, respectively. The other training recipes are the same as MiniLLM except batchsize=16 in the LLAMA-13B-7B pairs due to our limited computation resources.. We retrained all MiniLLM models with the official reproducible code. The metrics are averaged among 5 runs with 5 random seeds.

**Evaluation.** We evaluate image generation with common metrics: Frechet Inception Distance(FID), Inception Score (IS), Precision and Recall. The major metric is FID since it evaluates both diversity and fidelity. The results are evaluated with 250 DDPM sampling steps and 50000 samples generated with classifier-free guidance 3.8.

We evaluate LLM pretraining with SuperGLUE as the fine-tuning benchmarks. After pretraining with BabyLM dataset, the models are then finetuned with superGLUE. All the fine-tuning follows

the setting of BabyLLaMA to avoid overfitting. The metrics we used are the Matthews correlation coefficient (MCC), F1 score, and accuracy.

As for supervised fine-tuning tasks, we followed MiniLLM using 5 instruction-following datasets: DollyEval, SelfInst, VicunaEval, S-NI, and UnNI. Rouge-L score is the main metric for evaluating all models. It can measure the precision of the model's generation.

Our method is compared to previous method in Fig.A.

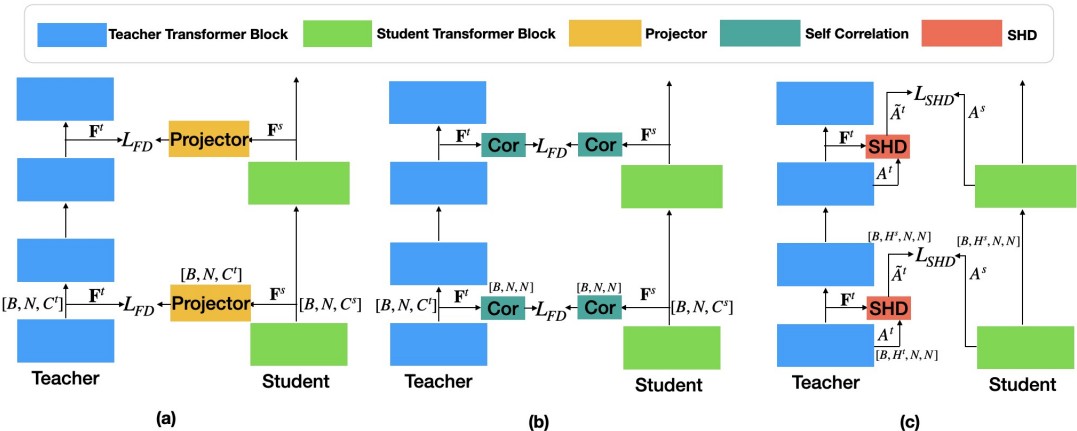

Figure 3: Comparison of typical representation distillation methods and our method. (a) Typical distillation uses a projector to align student features with teacher features, introducing extra parameters. (b) Typical distillation uses relations like self-correlations to align feature dimensions. (c) Our method, SHD, uses attention maps and outputs to squeeze attention maps, aligning with the student and ensuring minimal loss of knowledge transfer.

|  | Training Speed | Params |
|---|---|---|
| MiniLLM | 1.41s | 340M |
| MiniLLM+FD+Projector | 1.69s | 394M |
| MiniLLM+FD+Self_correlation | 1.49s | 340M |
| MiniLLM+VkD | 1.55s | 344M |
| MiniLLM+SHD | **1.41s** | **340M** |

Table 9: Training time cost of SHD and other FD-based methods with GPT2-Medium.

**Loss function selection.** We did ablation studies over loss function in Tab.10. KL performs better than MSE. We believe this is predictable since the same phenomenon happens in traditional logits supervision. And the performance always gets improved for whatever loss function is, proving SHD's positive impact.

| Loss Function | FID | IS | precision | Recall |
|---|---|---|---|---|
| KL | 36.95 | 46.27 | 0.5142 | 0.4843 |
| MSE | 38.04 | 44.34 | 0.5048 | 0.4921 |

Table 10: Comparison on different loss functions.

**Training speed of SHD.** Our method only computes the $\alpha$ in a batch-wise manner during training, which has a minor impact on the training speed of the baselines. We measured the training speed of MiniLLM and MiniLLM with SHD in Tab.9. We used 16 NVIDIA V100 GPUs to test the time. The time influence of SHD is negligible. Comparing to extra training parameters and extra training time like FD or VkD, the improvement of SHD is basically free.

**Distribution of $\alpha$ in SHD.** We sampled all training data (12.5k samples) from the Dolly dataset and calculate all $\alpha$ by $GPT2-XL$ to squeeze to 16 heads from 25 heads in total. The visualization of the $\alpha$ is in Fig.2. We can show that $\alpha$ always falls in the range of [0,1] for a pre-trained teacher model, making our method very stable in training, since it cannot provide any negative supervision of the attention maps. The distribution varies among different layers. Some certain heads like "Head 8" focus on one head in shallower layers and the other in deeper layers, indicating that our method can distinguish which head contains more useful information after squeezing and dynamically change the way of merging among layers.

**Complexity of SHD.** As illustrated in our method section, the complexity of SHD is similar to the attention mechanism, leading to a $O(N^2)$ complexity. Most intermediate computation are already done within original multi-head attention.

