# OpenReview forum: "Optimizing Knowledge Distillation in Transformers: Enabling Power of Multi-Head Attention without Alignment Barriers"
_ICLR.cc/2025/Conference — Submitted to ICLR 2025_

### Official Review · Reviewer_ARUU · 2024-10-31

**Soundness:** 3
**Presentation:** 2
**Contribution:** 3
**Rating:** 8
**Confidence:** 3

**Summary:**

The paper proposes a new method which tries to solve the issue of head misalignment in multi-head attention mechanism for knowledge distillation. The main idea of the paper is to compress multiple attention heads into a single one, and distill this knowledge. The authors show  experimental results in image generation, (small) LLM pretraining and finetuning, outperforming several other methods. Furthermore, they present an excellent collection of ablation studies, in regards to relation with other distillation models, main hyperparameters, alternatives to their method and the loss function.

**Strengths:**

The paper has the following strengths:

1) Motivation - the paper is naturally well-motivated, starting with some clear observations: i) the head redundancy in Transformers; ii) alignment missmatch where teacher and student have different dimensions (which they almost always do); 3) attention matrices being rank deficient. While all of these have been observed in the past, it is still a good way of formalizing them and then giving the solution by proposing the attention head compression using the linear optimization method (section 4.2).

2) The method provided here to merge the attention heads is elegant, mathematically justified, and to me makes sense. Its derivations is also quite complete and just.

3) The method is shown to improve over standard knowledge distillation models in different settings: Image generation, small LLM pretraining, and small LLM finetuning. The results in all cases are quite significant, convincingly outperform the baseline and in some cases are (almost) as good as the much larger teacher.

4) The ablation study at the end of the paper is excellent, showing the robustness of the method under different hyperparameter settings, comparison with other methods (hard selection, constant merging of heads) and comparing the KD loss with MSE.

**Weaknesses:**

I think the paper has some room for improvement:

1) While the paper explains how 2 attention heads can be merged, the paper mentions that multiple attention heads can be merged. That part is quite unclear to me and would be nice if the authors can clarify the following points:

a) How this is done, is it simply merging each pair of attention heads, thus reducing them by half, or if there is some iterative procedure.

b) If there is some iterative procedure, how this is done in the first place, is it done for several heads alltogether on in pairs. If for several heads, how does the procedure in 4.2 is extended, and furthermore do the alphas need to sum up to 1?

c) How are the heads to be selected chosen in the first place?

2) The size misalignment where the teacher has larger hidden dimensions than the student, is a real issue. Simply projecting the features of the teacher to smaller dimensions, or those of students to larger dimensions, usually destroys information making the KD process not work. However, there are some remedies of this that completely mitigate this issue, for example by using orthogonal projections, as the following paper, making the standard KD techniques work.

[A] Miles et al., VkD: Improving Knowledge Distillation using Orthogonal Projections, CVPR 2024.

While this paper and [A] target the problem quite differently, the motivations are quite similar, so it would be interesting if the paper can be discussed and compared with under some setting (image generation being the closest setting in them).

3) All the results are shown in generative tasks. It would be interesting to see if the method works in discriminative tasks (e.g., image classification and/or object detection), showing the method to be more general.

4) All the NLP results are shown in relatively small LLMs. It would be interesting to see if the method scales well to larger real-world LLMs. From my experience, this is always not the case, and what works in 'small' (tens to hundreds of millions of weights) does not neccesarily work in 'large' language models (billions of weights). If possible, it would be interesting to see how the method works when distilling the knowledge in LLama models. The ideal case would be Llama 13B -> Llama 7B, if that is not possible then something like Llama 7B -> TinyLlama 1.1B. Would be fine to use different alternatives to Llama (Mistral, Qwen-2), but it would be nice if they are a) relatively modern, b) relatively large.

5) Missing citation. The observation in equation 3 and 4, the idea that value-output multiplication can be considered as a single circuit (similar to query-key) has been done before, and it would be nice for the authors to cite it.

[B] Elhage et al., A Mathematical Framework for Transformer Circuits, Anthropic 2021, https://transformer-circuits.pub/2021/framework/index.html

6) Some clarification in Figure 1 is needed. If the results are from a single example, then that is very obviously expected. Of course that most attention heads are not going to produce anything meaningful (get 'activated'), that is by design. If not:

a) How did the authors choose the example?

b) Wouldn't it be better to check lots of inference examples, and merge the results?

7) Some writing issues:

a) Please rewrite the contributions in active instead of passive.

b) Figure 1 cannot be well read, and cannot be read at all if it is in printed paper. The authors should redesign it so it is more readable (probably by making it larger)

c) It is unclear to me what precision and recall means in the context of image generation (Table 1).

**Questions:**

I have already included several questions in the weaknesses. In particular:

1) The authors should ideally clarify the questions under weakness (1) and (6)

2) The authors should give credit to papers [A] and [B] and ideally compare with paper [A] under some setting.

3) If possible, the authors should compare their method on other settings: a) discriminative tasks; b) larger LLMs.

I am initially positive, the paper makes sense, seems to work in different settings, and reach good results. Thus, I am scoring it initially as (6) Borderline accept. However, I acknowledge that the paper has some issues with clarity, more comparisons, writing, and putting it in context of other works. Providing that the authors would well-address the mentioned issues, I will switch to (8) Accept.

---------------------------------------------------------------------------------------------------------------------------------------------------------------------------------------

After rebuttal:

I thank the authors for their excellent and detailed rebuttal.

1) I expressed some confusion about the merging process. The authors have adequately addressed this and provided additional results.

2) I mentioned VkD as missing reference and a recent paper that addresses a similar problem. The author have integrated the comparison with it in the manuscript, showing that their method outperforms VkD.

3) I mentioned lack of discriminative tasks. The authors have compared their method with other methods in a discriminative benchmark, showing that it outperforms the other methods.

4) I mentioned lack of experiments in real LLMs. The authors have shown an additional experiments in LLama13B-LLama7B setting.

5-6-7) The authors have adequately addressed my concerns about clarify, missing citation and metrics.

I was initially positive about the paper. After the rebuttal, the authors have adequately addressed all of my concerns, and have provided results that further improve the paper. Furthermore, checking the criticism from other reviewers, and the response by the authors, I do not think that the paper has any weakness that might warrant blocking the publications.

I think this is a very good to excellent paper. As promised, I am increasing my score to 8 and I am happy to further champion it.

---

> ### Author Response · Authors · 2024-11-20
> **Response to Reviewer ARUU**
>
> The authors sincerely thank the reviewer for the very complete and detailed comment. The suggestions of the reviewer are very helpful and insightful. We added and modified some contents in the paper as the reviewer suggested.
>
> **Regarding the Weakness 1:**
>
> We are sorry for the confusion we caused in the part of the merging process. As the reviewer said, we did the merging process progressively since each time of the merging process can make the number of heads halved. The result in Table 3 of the GPT2-XL teacher(25 heads) and GPT2-small student(12 heads) used SHD to merge heads progressively twice.
> | Method           | Head | Params | DollyEval | SelfInst | VincunaEval | S-NI | UnNI |
> |------------------|------|--------|-----------|----------|-------------|------|------|
> | Teacher(GPT2-XL) | 25   | 1.5B   | 27.6      | 14.3     | 16.3        | 27.6 | 31.8 |
> | SFT w/o KD       | 12   | 120M   | 23.3      | 10.0     | 14.7        | 18.5 | 18.5 |
> | KD               | 12   | 120M   | 22.8      | 10.8     | 13.4        | 16.4 | 22.0 |
> | MiniLLM          | 12   | 120M   | 24.6      | 13.2     | 16.9        | **25.1** | 25.6 |
> | MiniLLM+SHD      | 12   | 120M   | **24.8**      | **13.6**     | **18.0**        | **25.1** | **25.7** |
>
> We also did the experiments of matching heads with each other before training and computed the costs of head matching, so that the attention maps of merging heads have the highest similarity of attention maps on the total training dataset. We added the performance of head matching in Table 4.
> | Method                                       | DollyEval     | SelfInst      | VincunaEval   | S-NI          | UnNI          |
> |----------------------------------------------|---------------|---------------|---------------|---------------|---------------|
> | MiniLLM                                      | 25.4          | 14.6          | 17.7 | 27.4          | 31.3          |
> | MiniLLM+SHD                                  | **26.2**| **15.2** | **17.7**| **28.1** | **32.2** |
> | MiniLLM+SHD+head_matching | **26.3** | **15.3** | **18.2** | **28.1** | **32.3**|
> We want to emphasize that all other SHD experiments we reported, if not specified, did not use head matching since it requires evaluating the whole training dataset before training. We want our method to be plug-and-play and easy-to-follow.
>
>
> **Regarding the Weakness 2:**
>
> We reproduced VkD in the supervised fine-tuning tasks of LLM and migrated it from their official implementation as the reviewer suggested. We searched for the best hyper-parameter for VkD and reported the best result in the ablation study of Table 5. Please notice that our implementation uses fp32 to train VkD since the cuda version of the orthogonal projector in PyTorch only supports fp32. The performance of our SHD is reported in fp16 training and still has improvement over VkD. Besides, VkD is a simple and cost-effective constrained feature distillation pipeline. Our approach, however, is even more cheap—it's practically free.
> | Method                                       | DollyEval     | SelfInst      | VincunaEval   | S-NI          | UnNI          |
> |----------------------------------------------|---------------|---------------|---------------|---------------|---------------|
> | MiniLLM                                      | 25.4          | 14.6          | 17.7 | 27.4          | 31.3          |
> | MiniLLM+FD+Projector                         | 25.8          | **15.2** | 17.6          | 27.3          | 31.4          |
> | MiniLLM+FD+self_correlation                 | 25.9          | **15.2** | 15.8          | 26.8          | 31.7          |
> | MiniLLM+VkD               | 26.0          | 14.9          | **17.7** | 27.1          | 31.0          |
> | MiniLLM+SHD                                  | **26.2**| **15.2** | **17.7**| **28.1** | **32.2** |
> | MiniLLM+SHD+head_matching | **26.3** | **15.3** | **18.2** | **28.1** | **32.3**|
>
> **Regarding the Weakness 3:**
>
> We added the results of discriminative tasks as the reviewer suggested. We compared SHD with NKD, and ViTKD which focuses on the knowledge distillation of ViT-ViT teacher-student pairs. Our SHD still gains improvement whether independently or over ViTKD and shows compatibility with feature distillation and its generability.
> | Method                | Model       | Head | Epochs | Top1 Acc  |
> |-----------------------|-------------|------|--------|-----------|
> | Teacher               | DeiT3-small | 6    | 300    | 80.69     |
> | Baseline (without KD) | DeiT-Tiny   | 3    | 300    | 74.43     |
> | Baseline+SHD          | DeiT-Tiny   | 3    | 300    | **75.38** |
> | ViTKD+NKD             | DeiT-Tiny   | 3    | 300    | 77.79     |
> | ViTKD+NKD+SHD         | DeiT-Tiny   | 3    | 300    | **78.21** |
>
> The result of ViTKD+NKD+SHD also shows the compatibility of SHD with other FD methods, improving the performance of ViTKD+NKD by 0.42% on a strong baseline.

---

> ### Author Response · Authors · 2024-11-20
> **Response to Reviewer ARUU - part 2**
>
> **Regarding the Weakness 4:**
>
> Under limited computational resources, we added the result of Llama 13B-7B teacher-student pair in the SFT tasks. The batch size is set to 16 due to our limited computational resources. Our SHD surpasses the MiniLLM by 1.1% on UnNI. We showed the general ability over relatively large and relatively modern language models.
>
> | Teacher(LLaMA) | 40 | 13B | 29.7     | 23.4     | 19.4     | 35.8     | 38.5     |
> |---------------------|----|-----|----------|----------|----------|----------|----------|
> | MiniLLM             | 32 | 7B  | 28.9     | 23.1     | 19.4     | 34.8     | 37.4     |
> | MiniLLM+SHD         | 32 | 7B  | **29.1** | **23.4** | **20.0** | **34.9** | **38.5** |
>
> **Regarding the Weakness 5:**
>
> We are very sorry for the missing citations. We have added them to our citations.
>
> **Regarding the Weakness 6:**
>
> We chose the example randomly. We watched around 1000 inference samples and they all behave like this simple example. The redundancy in the behavior of heads is similar. We will add some visualized results of this phenomenon to our project site due to the limitation of pages of ICLR paper.
>
> **Regarding the Weakness 7:**
>
> We thank the reviewer for all the writing problems mentioned and we fixed them in the manuscript. Thanks so much for pointing them out.
> The precision and recall metrics are from [1]. The basic concept is similar to precision and recall in object detection.
>
> [1]:improved precision and recall metric for assessing generative models. NIPS2019

---

> > ### Comment · Reviewer_ARUU · 2024-11-20
> > **Thank you and updated my score**
> >
> > I thank the authors for a very strong rebuttal.
> >
> > I updated my score to 8. Please check detailed comments under Questions/After rebuttal.
> >
> > I hope the paper will be accepted to ICLR 2025 conference.

---

> > > ### Author Response · Authors · 2024-11-20
> > > **Thank Reviewer ARUU**
> > >
> > > The authors sincerely thank the reviewer for the fast and positive reply. Your comments truly help us make our paper better.
> > > Best wishes.

---

> ### Author Response · Authors · 2024-11-23
> **Thank the reviewer and please let us know if any further questions to raise the confidence.**
>
> The authors want to thank the reviewer for raising our score to 8 and admiring our revision again.
>
> We noticed that the reviewer's confidence score is still 3. Please let us know if there are any further questions we can answer to raise the confidence of the review. We are very happy to answer them.
>
> Best regards,
> Authors of Submission 1443.

---

### Official Review · Reviewer_Sz2t · 2024-11-03

**Soundness:** 2
**Presentation:** 3
**Contribution:** 2
**Rating:** 5
**Confidence:** 4

**Summary:**

This work investigates the knowledge distillation specially tailored for transformer architectures. Specifically, the authors study the misalignment of attention heads between the teacher and student. They propose the squeezing-head module to compress multiple attention heads into single one, alleviating the mismatch of attention heads between different architectures. More specifically, the authors formulate the attention heads as a convex hull and aim to seek a substitute with minimal reconstruction error. Experiments on image generation and natural language processing show effectiveness.

**Strengths:**

1. The motivation is important to KD in transformer architectures. Due to computation budget, different models vary from depth to width. This work proposes to address the problem of misalignment of attention heads between teacher model and student model without learnable projectors.
2. The experiments are thorough. The authors show competitive results of their method on various tasks and dataset.

**Weaknesses:**

1. I have some concerns about the formulation starting from line 258. The authors formulate the *random convex hull* among multiple attention heads (*vertices*) and aim to find a substitute with minimal reconstruction error through **convex combination**. The problem is that no evidence reveals and supports the linearity (Eq. 5, line 266)  between two attention maps. In other words, it is less meaningful if we use the linear combination of two attention maps to approximate the other attention map. As an alternative, I think PCA, optimal transport, and kernel method can be the better tools.
2. It seems like Eq. 9 (line 291) does not guarantee global minimum.
3. The experiments mainly focus on the heterogeneous architecture where two models have different attention heads. I wonder about the result of homogeneous architectures.

**Questions:**

1. If teacher and student has the same attention heads, can this approach consistently beats existing KD methods?
2. In addition to efficiency and well alignment, one advantage of multi-head attention over single-head attention is their descriptive power. How is Figure 1 after you combine multi-head into single head?

---

> ### Author Response · Authors · 2024-11-20
> **Response to Reviewer Sz2t**
>
> We want to thank the reviewer for the very detailed comment.
>
> **Regarding the Weakness 1:**
>
> The authors agree with the reviewer that PCA, optimal transport, and kernel methods can be better tools. However, all these tools are time-consuming for training a model, since we need to do it online. We need a practical and useful method to compute the supervision of attention maps. And convex combination is a great way of being practical and it can significantly improve the performance in our experiments as well.
>
> **Regarding the Weakness 2:**
>
> There seems to be a misunderstanding. It is the global minimum in the convex combination once the heads’ order to squeeze is determined. The merging coefficients in the range of [0,1] can guarantee the positiveness of attention maps, and they always fall into this range as we depicted in Fig. 2 in the appendix. We also reported the results of using the training dataset and teacher models to match teachers’ heads with each other as "head_matching". This should be the global minimum overall.
> | Method                                       | DollyEval     | SelfInst      | VincunaEval   | S-NI          | UnNI          |
> |----------------------------------------------|---------------|---------------|---------------|---------------|---------------|
> | MiniLLM                                      | 25.4          | 14.6          | 17.7 | 27.4          | 31.3          |
> | MiniLLM+FD+Projector                         | 25.8          | **15.2** | 17.6          | 27.3          | 31.4          |
> | MiniLLM+FD+self_correlation                 | 25.9          | **15.2** | 15.8          | 26.8          | 31.7          |
> | MiniLLM+VkD               | 26.0          | 14.9          | **17.7** | 27.1          | 31.0          |
> | MiniLLM+SHD                                  | **26.2**| **15.2** | **17.7**| **28.1** | **32.2** |
> | MiniLLM+SHD+head_matching | **26.3** | **15.3** | **18.2** | **28.1** | **32.3**|
>
> All other SHD experiments we reported, if not specified, did not use head matching since it requires evaluating the whole training dataset before training. We want our method to be plug-and-play and easy-to-follow.
>
> **Regarding the Weakness 3 and Question 2:**
>
> This is a practical problem that we always use the heterogeneous architecture where two models have different attention heads. Large teachers have large heads and small students have small heads in the knowledge distillation of transformers. Our work aims to transfer knowledge to train better models with smaller sizes.
> Also, our method aims to tackle the problem of misalignment of teacher heads and student heads. It does not include the situation where teacher and student have the same heads, since it's pretty rare.

---

> > ### Comment · Reviewer_Sz2t · 2024-11-23
> >
> > Dear authors,
> >
> > Thanks for the response. Most of my concerns are addressed.
> >
> > The only issue I can’t reconcile is the assumption of linearity among the attention heads (maps), which results in the convex combination described in your manuscript.
> >
> > Regards,

---

> ### Author Response · Authors · 2024-11-23
> **Response to Reviewer Sz2t**
>
> The authors thank the reviewer for telling us that most of the reviewer's concerns are addressed.
>
> We want to address the concern of the linear assumption from three terms: **1)the mathematical term, 2)the efficiency term, and 3)the effectiveness term.**
>
> ●**1)Mathematical Soundness:**
>
>  Convex combinations maintain the essential probabilistic properties of attention maps, which is crucial for effective knowledge distillation.While methods like PCA or kernel methods can capture correlations, they may not preserve the **non-negativity** and **row-sum-to-one** properties of attention maps. Therefore, **convex combination of merging coefficients between [0,1]** is mathematical important. We also showed that the distribution of coefficients we computed in SHD in the Appendix. The global minimum coefficients always fall into this range, proving our assumption to be rational.
>
> ●**2)Computational Efficiency:**
>
>  The linear complexity of our method ensures practicality and scalability, unlike computationally intensive alternatives like optimal transport. Our linear approximation algorithm operates with a complexity of O(N^2) whereas optimal transport solvers, such as Interior Point Methods, are computationally more expensive, requiring O(N^3） complexity, **which is unacceptable in practical training.**
>
> ●**3)Effectiveness:**
>
> We want to thank the reviewer for admiring the thoroughness of our experiments on four different tasks, including Image Classification (discriminative task), LLM SFT, LLM Pretraining, and Image Generation (generative tasks), where SHD shows consistent improvement over baselines. We also compared with projector-based methods, which can be seen as a variant type for kernel methods, and outperformed these projector-based methods both on training speed and performance.The effectiveness of the linear assumption is further proved by these experiments.
>
> **While PCA, optimal transport, and kernel methods have their own merits, they introduce complexities and limitations that make them less suitable for our specific context according to the explanations above. We believe that our approach strikes an optimal balance between theoretical rigor, practical efficiency, effectiveness, and interpretability.**
>
> **We sincerely and kindly ask the reviewer to reconcile based on the explanation above since it is the only issue as the reviewer mentioned.** We are open to any further requirements by the reviewer.

---

### Official Review · Reviewer_ZNbn · 2024-11-03

**Soundness:** 3
**Presentation:** 2
**Contribution:** 2
**Rating:** 5
**Confidence:** 4

**Summary:**

This paper examines knowledge distillation in transformer architectures. Based on three observations in transformer attention maps, the proposed method investigates attention compression through linear approximation, using the squeezed attention maps for knowledge transfer between teacher and student models. Experiments are conducted on image generation and language model training tasks to demonstrate the effectiveness of the approach.

**Strengths:**

- This paper addresses a critical challenge in the era of large language models (LLMs): compressing transformer models using knowledge distillation techniques.

- The proposed method is easy-to-follow.

**Weaknesses:**

- Observations on transformer attention maps lack clarity and sufficient justification. For instance, the statement that “the number of heads is redundant to some degree” is not well-supported.

- The experiments do not adequately demonstrate the effectiveness of the proposed method. Key baseline and comparison methods are missing, which makes it difficult to evaluate the method’s impact on improving knowledge distillation in transformer architectures.

**Questions:**

See weakness.

---

> ### Author Response · Authors · 2024-11-20
> **Response to Reviewer ZNbn**
>
> We want to thank the reviewer for the comment.
>
> **Regarding the Weakness 1:**
>
> The observations that the reviewer referred to were mentioned in previous papers[1] and are not what we mainly discussed in the paper. Our paper mainly focuses on our method: SHD.
> [1] Are Sixteen Heads Really Better than One? (NIPS2019)
>
> **Regarding the Weakness 2:**
>
> We added the comparison between SHD, ViTKD, NKD, and V_kD, FD with projector and FD with self_correlation.
> | Method                                       | DollyEval     | SelfInst      | VincunaEval   | S-NI          | UnNI          |
> |----------------------------------------------|---------------|---------------|---------------|---------------|---------------|
> | MiniLLM                                      | 25.4          | 14.6          | 17.7 | 27.4          | 31.3          |
> | MiniLLM+FD+Projector                         | 25.8          | **15.2** | 17.6          | 27.3          | 31.4          |
> | MiniLLM+FD+self_correlation                 | 25.9          | **15.2** | 15.8          | 26.8          | 31.7          |
> | MiniLLM+VkD               | 26.0          | 14.9          | **17.7** | 27.1          | 31.0          |
> | MiniLLM+SHD                                  | **26.2**| **15.2** | **17.7**| **28.1** | **32.2** |
> | MiniLLM+SHD+head_matching | **26.3** | **15.3** | **18.2** | **28.1** | **32.3**|
>
> | Method                | Model       | Head | Epochs | Top1 Acc  |
> |-----------------------|-------------|------|--------|-----------|
> | Teacher               | DeiT3-small | 6    | 300    | 80.69     |
> | Baseline (without KD) | DeiT-Tiny   | 3    | 300    | 74.43     |
> | Baseline+SHD          | DeiT-Tiny   | 3    | 300    | **75.38** |
> | ViTKD+NKD             | DeiT-Tiny   | 3    | 300    | 77.79     |
> | ViTKD+NKD+SHD         | DeiT-Tiny   | 3    | 300    | **78.21** |

---

### Official Review · Reviewer_up1V · 2024-11-05

**Soundness:** 2
**Presentation:** 2
**Contribution:** 2
**Rating:** 5
**Confidence:** 4

**Summary:**

The paper proposes a new feature-based knowledge distillation method, specially tailored to Transformers. Specifically, the method reduces the number of attention maps to any desired number through linear approximation, without requiring additional projectors or parameters. The experiments are conducted on both vision tasks and language tasks, validating the effectiveness of the method.

**Strengths:**

1. The method is straightforward and reasonable, which is easy to understand.
2. The experiments are performed on both vision tasks and language tasks.

**Weaknesses:**

1. Fundamentally, the paper criticizes previous feature distillation methods because they often require projectors to align features (or special modifications to the model architecture), which introduce additional training costs. However, to the best of my knowledge, the prevalent of feature projectors (such as linear projectors, conv projectors, and MHA projectors) are rather lightweight, compared with the overhead of the teacher model and the student model.  The paper does not provide the FLOPs and training speed comparison between FD and SHD to show the necessity of designing SHD.

2. Furthermore, although the paper criticizes previous FD methods, it only compares with the most basic FD (with the basic linear projector) in the ablation study. The paper should cover the performance comparisons with various Top FD methods to enhance this claim.

3. In these experiments, it seems that SHD must be applied with logit-based methods and the improvement seems rather limited.  I wish to see the results that SHD independently applied to the network pair.


3. Although SHD does not have special designs for generative tasks,  all experiments are performed on image-generative tasks and text-generative tasks. I wish to see more results on image classification tasks, which are feasible to make comparisons with various Top FD methods.

**Questions:**

See weakness.

---

> ### Author Response · Authors · 2024-11-20
> **Response to Reviewer up1V**
>
> We thank the reviewer for the thorough response.
>
> **Regarding the Weakness 1:**
>
> As the reviewer said, the feature projectors in the knowledge distillation of image classification are quite lightweight since most FD methods only focus on a single layer of features in the backbone network (mostly features of the last layer before the classification head) due to the limitation of computational costs and performance.
> We also want to mention that most FD methods experimented on CNN-CNN/CNN-ViT teacher-student pairs in the image classification or detection tasks. When it comes to generative models, some research has suggested that the supervision of FD from each layer is a better solution in the training [1]. Our ablation studies between FD and SHD are computed on all layers of the student model.
> The training parameters and the training speed of FD, VkD[2], and SHD are reported and added to our new manuscript.
> Here is the result:
> |                              | Training Speed | Params        |
> |------------------------------|----------------|---------------|
> | MiniLLM                      | **1.41s**          | **340M**          |
> | MiniLLM+FD+Projector         | 1.69s          | 394M          |
> | MiniLLM+FD+Self\_correlation | 1.49s          | 340M          |
> | MiniLLM+VkD                  | 1.55s          | 344M          |
> | MiniLLM+SHD                  | **1.41s** | **340M** |
>
> Here we present an overall result of FLOPs. Consider a training scenario where the batch size is denoted by $b$, the hidden layer dimension by $h$, and the sequence length by $s$. The total FLOPs for the self-attention and MLP layers during both the forward and backward propagation processes for each layer is given by $(24bsh^2 + 4bsh^2) \times 3$. Taking the linear projector as an example, the additional computational cost is $4bsh^2 \times 3$. Our method does not incur any extra overhead for backpropagation. The additional computational cost introduced by our method is $2bsh^2$. Therefore, the proportion of additional computation due to the linear projector (relative to the student network) is $\frac{1}{6 + \frac{s}{h}}$, and the proportion of additional computation due to our method is $\frac{1}{6 + \frac{36h}{s}}$.
> Taking $h = 1024$ and $s = 512$ as an example, the proportion of additional computation due to the linear projector relative to the original model is **15.38\%**, while our method is approximately **1.28\%.**
>
> [1]Patient Knowledge Distillation for BERT Model Compression, EMNLP2019
> [2]VkD:  Improving Knowledge Distillation using Orthogonal Projections, CVPR 2024
>
> **Regarding the Weakness 2:**
>
> As Reviewer ARUU suggested, we further compared  VkD (CVPR 2024) with SHD, one of the state-of-the-art methods in FD. Our method outperforms VkD on all five test datasets.
>
> Also, we want to mention that SHD is not incompatible with most FD methods. The further discriminative experiments we added to the manuscript have clarified that. We added SHD to ViTKD[3], which mainly focuses on distilling features for ViT-ViT teacher-student pairs. The SHD can still improve the accuracy with a strong FD baseline.
> | Method        | Model       | Head | Epochs | Top1 Acc       |
> |---------------|-------------|------|--------|----------------|
> | Teacher       | DeiT3-small | 6    | 300    | 80.69          |
> | ViTKD+NKD     | DeiT-Tiny   | 3    | 300    | 77.79          |
> | ViTKD+NKD+SHD | DeiT-Tiny   | 3    | 300    | **78.21** |
>
> [3]ViTKD: Practical Guidelines for ViT feature knowledge distillation
>
> **Regarding the Weakness 3&4:**
>
> As mentioned before, we added discriminative experiments to our paper thanks to all reviewers.
> We did the single ablation study without logits-based KD in the discriminative tasks to show that our SHD can still improve performance significantly.
>
> | Method                | Model       | Head | Epochs | Top1 Acc       |
> |-----------------------|-------------|------|--------|----------------|
> | Teacher               | DeiT3-small | 6    | 300    | 80.69          |
> | Baseline (without KD) | DeiT-Tiny   | 3    | 300    | 74.43          |
> | Baseline+SHD          | DeiT-Tiny   | 3    | 300    | **75.38** |
>
>
> The reason why we reported performance with logits-kd methods for most experiments is that we want to show that even strong baselines can benefit from SHD and logits-kd methods are mostly near-cost-free. We did experiments on image classification, image generation, LLM pretraining, and LLM SFT, on which SHD shows consistent improvement.

---

### Author Response · Authors · 2024-11-20
**General response to all reviewers**

We greatly appreciate the reviewers' detailed responses and insightful suggestions. Your reviews truly help us polish our paper better. We sincerely appreciate that all reviewers believed our method is easy-to-follow,  straightforward and our experiments are thorough.  Additionally, Reviewer  ARUU affirmed our motivation and mathematics, improvement in all kinds of experiment settings, and the completeness of our ablation studies.  Reviewer Sz2t also appreciates the importance of solving the problem of misalignment of attention heads between the teacher and the student without learnable projectors. We will respond to each reviewer's questions and comments, respectively. We also revised our manuscript and updated it according to the reviewers' suggestions. Major modifications are highlighted in blue color.

---

### Author Response · Authors · 2024-11-23
**General response to all reviewers**

Dear Reviewers,

The authors sincerely appreciate the time and effort you are dedicating to reviewing our paper, "Optimizing Knowledge Distillation in Transformers: Enabling Power of Multi-Head Attention without Alignment Barriers".

As the rebuttal phase progresses, we wanted to kindly follow up on the status of your feedback. One of the reviewers **ARUU** has already shared detailed and encouraging comments, which we are truly grateful for. The reviewer **ARUU** thinks that our revision addresses the problems of clarity, more comparisons, writing, and putting it in context of other works. **The reviewer ARUU also believes that “our paper has no any further weakness that might warrant blocking the publication” by checking the criticism from other reviewers and the response by the authors.**

We cannot be more thankful for all the precious comments. We would greatly value your input as well to ensure a thorough and balanced evaluation of our work.

**Given the approaching deadline for final decisions, we kindly ask if it would be possible to share your feedback at your earliest convenience. Please do not hesitate to let us know if there are specific points we can clarify or address further.**

Thank you again for your valuable contributions to this process.

Best Wishes,

Authors of Submision 1443

---

### Author Response · Authors · 2024-12-02
**General response to all reviewers**

The authors want to thank the reviewers for the efforts in the reviews.
As the deadline of rebuttal disccusion is approaching, we kindly ask if it would be possible to share your feedback at your earliest convenience. Please do not hesitate to let us know if there are specific points we can clarify or address further. The authors have added promising results and replied to the concerns and weaknesses. We are confident with our method and experiments results and williing to answer any further questions.

We believe your suggestions and our efforts get our paper improved and meet the high standard of ICLR.

Best regards,

Authors of submission 1443

---

### Author Response · Authors · 2024-12-02
**General response to all reviewers**

Rebuttal Summary
We sincerely thank all the reviewers for their detailed and constructive feedback,**especially reviewer ARUU, who has the most insightful suggestions for the papers, the fastest responses to our rebuttal, and the most paper-related questions. The reviewer ARUU also believes that “our paper has no any further weakness that might warrant blocking the publication” by checking the criticism from other reviewers and the response by the authors.** Below, we summarize our key responses to address the main weaknesses raised by the reviewers.
***

## **Regarding Weakness 1: On computational efficiency and feature distillation (FD)**



●Feature Projectors: We clarified that our ablation studies involve comparisons across all layers of the student model, contrasting FD with VkD [2] and our method, SHD. Results show that SHD achieves comparable efficiency while avoiding the overhead of projectors and enhancing training speed and parameter efficiency.

●Computational Complexity: We provided a detailed analysis demonstrating that SHD introduces significantly less computational overhead (1.28%) compared to linear projectors (15.38%).

●Mathematical Justification: Convex combinations, a foundation of SHD, ensure attention map properties (non-negativity, row-sum-to-one) while maintaining computational practicality and theoretical rigor.

***

## **Regarding Weakness 2: Compatibility with other methods and performance comparisons and Lack of discriminative tasks**

**We added SHD results on image classification of ImageNet-1k.**

●State-of-the-Art Comparisons: SHD was compared with VkD [2] and ViTKD [3] across tasks like image classification, LLM pretraining, and supervised fine-tuning, **consistently outperforming these baselines.**

●Independent Effectiveness: Ablation studies showed SHD improves performance even without logits-based KD, proving its standalone effectiveness (e.g., **+0.95%** Top-1 accuracy on DeiT-Tiny).

●Compatibility: We demonstrated SHD's compatibility with other feature distillation methods, showing it improves performance even when combined with strong baselines e.g. FD methods and logits-KD methods, achieving a **0.42%** improvement over NKD+ViTKD, **proving our compatibility with traditional FD methods and logits-KD methods.**

***

## **Regarding Weakness 3: Scalability**

●Scalability: SHD showed superior generality with modern, large-scale language models (e.g., LLaMA 13B → 7B), outperforming baselines under constrained computational resources.

***

## **Regarding Weakness 4: Linear Assumption in SHD**

We addressed the concern of the linear assumption from three terms: **1)the mathematical term, 2)the efficiency term, and 3)the effectiveness term.**

**Mathematical Soundness**: Our method's use of convex combinations ensures that the probabilistic properties of attention maps are preserved, unlike PCA or kernel methods which may not maintain non-negativity and row-sum-to-one properties. The coefficients used in our method are mathematically justified, as they always fall within the [0,1] range, which is crucial for effective knowledge distillation.

**Computational Efficiency**: Our method offers linear complexity (O(N^2)), making it practical and scalable compared to more computationally demanding alternatives like optimal transport, which requires O(N^3) complexity. This efficiency is a significant advantage in practical training scenarios.

**Effectiveness**: The effectiveness of our method is demonstrated through extensive experiments across four different tasks, where our method (SHD) consistently outperforms baselines and projector-based methods in both training speed and performance. These results validate the effectiveness of our linear assumption and show that our approach offers a superior balance between theoretical rigor, practical efficiency, effectiveness, and interpretability compared to PCA, optimal transport, and kernel methods.


***

## **Regarding Weakness 5: Missing citations, Writing issues**

●We addressed the missing citations and added references to ensure proper acknowledgment of prior work. We also revised the manuscript thoroughly to address all writing concerns.

***

# **Conclusion and Final Appeal**

**We are pleased to confirm that all issues raised by the reviewers have been fully addressed through extensive experiments, theoretical analysis, and clarifications. The revisions strengthen the manuscript significantly, substantiating SHD's practicality, robustness, and effectiveness across diverse tasks.**

We deeply appreciate the reviewers’ thoughtful feedback and respectfully request them to kindly reconsider their initial scores in light of the comprehensive updates. **We believe the resolved issues, coupled with our clarified contributions and rigorous evaluations, warrant a positive reassessment of our work.**

We remain open to further questions or suggestions and sincerely thank the reviewers for reviewing our submission.

---

### Meta-Review · Area_Chair_7PZf · 2024-12-18

**Metareview:**

The paper proposes a distillation method for transformer-based models. The main idea is to reduce the number of attention maps to any desired number through linear approximation, which is simple but effective. However, the authors have not sufficiently demonstrated why the linear assumption is valid or clearly articulated the motivation and advantages of the method. Additionally, it would be beneficial to illustrate the main idea with figures to enhance understanding. Based on these strengths and weaknesses, the decision is not to recommend acceptance at this time. We encourage the authors to carefully consider the reviewers' comments when revising the paper for submission elsewhere, particularly in appropriately articulating the motivation and advantages of the method.

**Additional Comments On Reviewer Discussion:**

The paper was reviewed by four experts in the field and finally received diverse scores: 8, 5,5, and 5. The single positive score of 8 is given with low confidence.
The major concerns of the reviewers are:
1.	the linear assumption for attention maps,
2.	the effectiveness of the method for homogeneous architectures,
3.	the improvements for some experimental settings are marginal.

The authors failed to address the above concerns during the discussion period.
I fully agree with these concerns and, therefore, make the decision to reject the paper.

---

### Decision · Program_Chairs · 2025-01-22

Reject